# A Postmortem Case Study—An Analysis of microRNA Patterns in a Korean Native Male Calf (*Bos taurus coreanae*) That Died of Fat Necrosis

**DOI:** 10.3390/ani13132149

**Published:** 2023-06-29

**Authors:** Sang-Joon Lee, Ho-Seong Cho, Sanghyun Noh, Young Hun Kim, Hwi-Won Seo, Yeonsu Oh

**Affiliations:** 1College of Veterinary Medicine and Institute of Veterinary Science, Kangwon National University, Chuncheon 24341, Republic of Korea; sjoon516@kangwon.ac.kr (S.-J.L.); livensh@gmail.com (S.N.); 2College of Veterinary Medicine and Bio-Safety Research Institute, Jeonbuk National University, Iksan 54596, Republic of Korea; hscho@jbnu.ac.kr; 3Division of Companion Animal Science, Woosong Infomation College, Daejeon 34606, Republic of Korea; 86vetlover@naver.com; 4Infectious Disease Research Center, Korea Research Institute of Bioscience and Biotechnology, Daejeon 34141, Republic of Korea; seohw01@kribb.re.kr

**Keywords:** bovine fat necrosis, diabetes mellitus, serum microRNA

## Abstract

**Simple Summary:**

A 6-month-old male Korean native calf, known for its highly marbled meat, was purchased but found dead three days later, without any notable medical history. The calf had a significantly higher body weight than average for its age. Postmortem examination revealed fat necrosis, along with gastric dilatation, volvulus and intestinal obstruction, as the cause of death. Serum chemistry analysis indicated disrupted lipid metabolism and pancreatic damage—obstructive acute pancreatitis. To investigate the cause of fat necrosis, microRNA patterns were analyzed. Interestingly, the results resembled those found in human patients with diabetes mellitus. To explore the potential correlation between fat necrosis and human diabetic-like miRNA patterns, a comprehensive analysis of miRNAs and the miRNA-mediated Kyoto Encyclopedia of Gene and Genome (KEGG) pathway were conducted. The bovine target gene prediction and KEGG analysis indicated a significant association between multiple genes and diabetic-like clinical conditions. These findings suggest fat necrosis can occur in high-spec cattle such as Korean native calves, and in such cases clinical indicators such as diabetes are observed, providing the factors contributing to the breed’s high meat quality and marbling. The in-depth analysis of miRNA and the KEGG pathway offers valuable insights into the potential mechanisms underlying this correlation.

**Abstract:**

Korean native cattle are highly valued for their rich marbling and flavor. Nonetheless, endeavors to enhance marbling levels can result in obesity, a prevalent contributor to fat necrosis. Fat necrosis is characterized by the formation of necrotic fat masses in the abdominal cavity, which physically puts pressure on affected organs, causing physical torsion or obstruction, resulting in death and consequent economic loss. Pancreatic injuries or diabetes mellitus were reported as factors of fat necrosis in humans; however, the pathogenesis in animals has not been established. In this study, we identified fat necrosis in a 6-month-old Korean native cow and investigated its potential underlying causes. Serum samples were utilized for a microarray analysis of bovine miRNA. Comparative examination of miRNA expression levels between cattle afflicted with fat necrosis and healthy cattle unveiled notable variances in 24 miRNAs, such as bta-miR-26a, bta-miR-29a, bta-miR-30a-5p and bta-miR-181a. Upon conducting miRNA-mediated KEGG pathway analysis, several pathways including the prolactin signal pathway, insulin resistance, autophagy, the insulin-signaling pathway and the FoxO-signaling pathway were found to be significantly enriched in the calf affected by fat necrosis. As a result, this study potentially indicates a potential connection between fat necrosis and diabetes in Korean native cattle.

## 1. Introduction

Since the domestication of cattle began over 8000 years ago, cattle have adapted to diverse geographical regions and various breeding purposes [1]. Approximately 1600 breeds are registered with the Food and Agriculture Organization of the United Nations (FAO), and 120 breeds are being raised and conserved worldwide for livestock. Among them, there are more than 16 breeds of cattle originating in Asia, such as Wagyu from Japan and Sahiwal from India [2]. These breeds possess unique characteristics influenced by factors such as climate and genetic variations.

Korean native cattle (*Bos taurus coreanae*) hold a prestigious status and are extensively bred across the Republic of Korea. Within the beef industry of South Korea, these cattle have garnered notable recognition for their high exceptional marbling content and superior meat quality characteristics. Consequently, producers have prioritized the augmentation of intramuscular fat to enhance flavor and have established higher marbling score requirements to reap greater economic advantages. However, the pursuit of elevated marbling scores, accomplished through genetic selection and feeding strategies, renders them susceptible to obesity, a prominent contributing factor to the occurrence of fat necrosis [3].

Fat necrosis is a pathological phenomenon defined by the demise of adipose tissue and is frequently encountered in diverse organs. In the realm of veterinary medicine, it manifests in various mammalian species, with a notable prevalence in bovines, where it is distinguished by the development of necrotic fat masses within the abdominal cavity [4,5,6]. The visceral fat of several kinds of domestic animals and of some wild animals occasionally develops disseminated lesions seen grossly as yellowish-white or white patches and nodules [5]. These necrotic fat masses are multifocally found in various sites, including intestines, mesentery of the spiral colon, mesorectum and the retroperitoneal area. This disease is very complex and causes intestinal obstruction and some clinical symptoms, such as sclerous feces, constipation and chronic anorexia [7]. Cattle afflicted with fat necrosis exhibit digestive disruptions caused by physical constriction or obstruction from the fat masses. Symptoms include reduced fecal output, bloating and, in some cases, mortality [6]. The incidence of fat necrosis has been reported in the cattle industry since the 1960s, especially in Japan [7] and Korea [4]. Obesity, heredity, lack of exercise and intake of a high-energy diet have all been associated with the occurrence of fat necrosis [7]. With the transition of Korean native cattle breeding towards intensive fattening management measures, it has been reported that the occurrence of fat necrosis is progressively rising, and the age of onset is decreasing. Consequently, the economic repercussions stemming from fat necrosis continue to expand [4]. The exact pathogenesis of fat necrosis is still unknown, but it has been documented that pancreatic injuries such as acute pancreatitis and acute pancreatic necrosis are involved in the formation of necrotic fat lesions and systemic inflammatory response in bovine [7].

MicroRNAs (miRNAs) are specialized short non-coding RNAs that play critical roles in post-transcriptional gene regulation inhibiting target mRNA translation. Recent research on circulating miRNAs highlights their usefulness as biomarkers of diseases. Under various conditions, cells release miRNAs that are free or in microvesicles that can be taken up by other cell types. These extracellular miRNAs are important mediators of cell-to-cell communications and coordinate biological functions including angiogenesis, tumor cell invasion and immune response [8]. Given their abundant and tissue-specific expression, as well as their resistance to degradation in serum, miRNAs can serve as valuable biomarkers for monitoring a range of diseases, utilizing a straightforward blood draw [8,9]. Additionally, miRNAs are widely conserved across different species, further enhancing their potential as diagnostic indicators. Emerging research has indicated that miRNAs play pivotal roles in various cellular processes, encompassing differentiation, proliferation, apoptosis and immune response. Furthermore, dysregulation of miRNAs has been observed across numerous diseases, highlighting their significance in pathological conditions [9]. Moreover, the utilization of miRNA-mediated Kyoto Encyclopedia of Gene and Genome (KEGG) pathway analysis has emerged as a promising method to explore the involvement of miRNAs in diverse biological processes and diseases [10,11]. Through the integration of miRNA target predictions and KEGG pathway data, this approach enables identification of pathways regulated by miRNAs.

In our current investigation, we conducted an extensive postmortem pathological examination on an obese Korean native calf (*Bos taurus coreanae*), uncovering extensive fat accumulation accompanied by fat necrosis. Subsequently, we compared the alterations in miRNAs between healthy Korean native cattle and the affected calf. Our objective was to probe into the potential underlying factors contributing to fat necrosis through miRNA-mediated KEGG pathway analysis.

## 2. Materials and Methods

### 2.1. Animal, History and Postmortem Examination

A male Korean native calf (*Bos taurus coreanae*), aged 6 months and weighing 250 kg, was discovered deceased on a farm. Despite being purchased only three days prior to its demise, no notable clinical history was reported by the farm owner. A postmortem examination was conducted on site using standard procedures, and any abnormalities were documented. Following the necropsy, tissue samples were collected and preserved in 10% neutral buffered formalin for subsequent histopathological analysis.

### 2.2. Histopathology

The tissue samples fixed in formalin were trimmed and subjected to dehydration through graded ethanol and cleating in xylene, utilizing a HistoCore PEARL tissue processor (Leica Biosystems, Wetzlar, Germany). Subsequently, the specimens were embedded in paraffin blocks. Formalin fixed paraffin embedded (FFPE) tissues were then sectioned into 4 μm and stained in hematoxylin and eosin (H&E).

### 2.3. Serum Chemistry

The blood samples were taken as soon as possible from a cadaver after being found dead. Two types of serum chemistry analysis were conducted to check the accuracy and the reproducibility of the results to see how accurate the dead blood sample is: wet serum chemistry and dry serum chemistry. Both methods showed the same results in the chosen parameters. Serum levels of amylase, lipase, alkaline phosphatase (ALP) and triglycerides were chosen to evaluate the typical pancreatic serum chemistry profiles. These parameters were analyzed by Neodin Biovet Laboratory (Guri, Republic of Korea).

### 2.4. MiRNA Extraction and Analysis

Serum samples were collected from a commissioned bovine and two healthy bovines, ensuring they shared the same species, age and sex for miRNA comparison. Each sample was duplicated for statistical analysis. MiRNAs were extracted from the serum samples using miRNeasy serum/plasma kit (Qiagen, Hilden, Germany) following the manufacturer’s protocol. Subsequently, cDNA synthesis was performed using the TaqMan^®^ MicroRNA Reverse Transcription Kit (Applied Biosystems, CA, USA) as per the manufacturer’s instructions. After conducting quality-control measures, a comprehensive profiling of 783 bovine miRNAs was carried out using the Affymetrix GeneChip miRNA 4.0 array (Affymetrix, CA, USA), following the manufacturer’s guidelines at Macrogen (Guri, Republic of Korea).

### 2.5. Raw Data Preparation

The raw data were automatically extracted using the Affymetrix GeneChip^®^ Command Console^®^ software version 4.0 (Affymetrix, CA, USA) following their data extraction protocol. Subsequently, the CEL files were imported, and miRNA level RMA + DABG-All analysis was performed using the Affymetrix^®^ Power Tools software version 2.10.0 (Affymetrix, CA, USA). The array data were filtered based on the probes annotated for the species of interest.

### 2.6. Target Gene Prediction and Pathway Analysis

To identify miRNA-targeted genes, established methodologies previously described [12] were employed. TargetScan (http://www.targetscan.org; accessed on 30 March 2023) was utilized, with selection criteria of cumulative weighted value < −0.4 for Targetscan. The target genes that were commonly identified were then subjected to gene ontology (GO) annotation and enrichment analysis across three ontologies: molecular function, cellular component and biological process. This analysis was performed using the WEB-based gene set analysis toolkit (Web-Gestalt) (http: //www.webgestalt.org/#; accessed on 30 March 2023, version 2019). Furthermore, pathway analysis was conducted using the Kyoto Encyclopedia of Gene and Genome (KEGG). A threshold criteria of false discovery rate (FDR) < 0.05 was applied for determining significant associations in the pathway analysis.

### 2.7. Statistical Analysis

For the comparative analysis of miRNA between the test sample and control sample, an independent *t*-test and log2 fold change were employed. The null hypothesis assumed no differences among the groups. To control the false discovery rate (FDR), *p*-values were adjusted using the Benjamini–Hochberg algorithm. All statistical tests and visualizations of the differentially expressed genes were conducted using R statistical language version 3.3.2 (R Foundation for Statistical Computing, Vienna, Austria).

## 3. Results

### 3.1. Gross Pathological Findings

At necropsy, there were large amounts of hard nodular accumulations of fat surrounding and infiltrating many of the abdominal visceral organs and support structures. The mesentery surrounding the small intestines and colon was infiltrated with abundant firm fat that compressed intestines and filled greater than 50% of the peritoneal cavity. The retroperitoneal space was also distended with firm fat, which was mottled white, tan and golden-brown, and was often gritty when sectioned. Approximately 80% of the jejunum and 90% of the duodenum had diffuse transmural reddening, and half of the abomasum was edematous and reddened transmurally. The hard fat nodules firmly surrounded numerous systemic organs, including mesentery (Figure 1A), lung (Figure 1B), heart (Figure 1C) and predominantly the kidney (Figure 1D). These fat tissues infiltrated the renal sinus and interlobar spaces of the kidney. While the thickness of the fat masses measured approximately 1 cm in several organs, the ones present on the kidney exhibited a diameter of about 3 cm, with the thickest mass reaching approximately 8 cm. Furthermore, gastric dilation and volvulus (GDV), intestinal torsion characterized by diffuse dark red discoloration and a colon filled with blood were also observed within the digestive system. These findings suggest that the pronounced fat accumulation throughout the mesentery physically impacted these organs and could be considered as potential causes of death.

### 3.2. Histopathological Findings

Microscopic examination of peritoneal fat confirmed necrosis of large areas of fat and the thick, hard and firm fat nodules enveloping the organs. The perirenal area revealed infiltration and partial effacement by adipose tissue (Figure 2A). When viewed under high-power magnification, the adipocytes exhibited indistinct cellular outlines, a scarcity of peripheral nuclei and an elevated presence of blood vessels, which are characteristic features of fat necrosis (Figure 2B). Consistent with the gross observations, other histopathological findings were in line with these observations.

### 3.3. Serum Chemistry

The serum chemistry analysis of the calf revealed notable abnormalities, including hypertriglyceridemia, hyperamylasemia, hyperlipasemia and elevated alkaline phosphatase (ALP) levels. Specifically, the serum triglyceride levels were measured at 407 mg/dL, which exceeds the normal range of 10–19 mg/dL. Additionally, the calf exhibited elevated levels of amylase (75.4 U/L, normal range: 14–50 U/L), lipase (19.5 U/L, normal range: 5–13 U/L) and ALP (353 U/L, normal range: 27–127 U/L). These findings further support the possibility that the calf is experiencing fat necrosis, potentially linked to obesity and pancreatic injury [13].

### 3.4. Analysis of miRNA Expression Pattern and KEGG Pathway

Out of the completed set of 783 miRNAs in bovines, a total of 72 miRNAs displayed notable differences between the compared groups. Building upon previous research, it was identified that 24 out of these 72 miRNAs are known to be linked with diabetes mellitus, prompting their inclusion for further analysis [14,15,16,17]. Upon comparing healthy animals with those affected by fat necrosis, it was observed that 22 bovine miRNAs exhibited significant upregulation, while 2 bovine miRNAs displayed significant downregulation in the animals with fat necrosis (Table 1).

Utilizing the Morpheus web-based tool (https://software.broadinstitute.org/morpheus/; accessed on 30 March 2023), we generated a heat map accompanied by hierarchical clustering to visualize the expression patterns of 24 miRNAs (Figure 3). The hierarchical clustering analysis illustrated the differential expression of miRNAs between the upregulated and downregulated miRNAs. Subsequently, a total of 838 target genes associated with the upregulated bovine miRNAs in the fat necrosis samples were identified through TargetScan. These target genes were then subjected to KEGG pathway analysis using the Web-Gestalt webserver. The analysis revealed several significantly enriched pathways (FDR < 0.05) targeted by upregulated miRNAs in the fat necrosis sample, including the prolactin-signaling pathway, insulin resistance, autophagy, the insulin-signaling pathway, the FoxO-signaling pathway, focal adhesion, microRNAs in cancer, the PI3K-Akt-signaling pathway and pathways in cancer (Table 2). The downregulated miRNAs in the fat necrosis calf were found to target several pathways, including fatty acid biosynthesis, folate biosynthesis, bacterial invasion of epithelial cells, the PPAR-signaling pathway, cardiac muscle contraction, hypertrophic cardiomyopathy, dilated cardiomyopathy, the TNF-signaling pathway, signaling pathways regulating the pluripotency of stem cells and ubiquitin-mediated proteolysis. However, it is important to note that these downregulated pathways did not exhibit statistical significance (FDR > 0.05).

## 4. Discussion

Fat necrosis accompanied with lipomatosis is often an incidental finding but has been purported to cause intestinal obstruction and compression of other structures resulting in death [6]. In the current study, we conducted a thorough postmortem pathological examination of an obese Korean native calf (*Bos taurus coreanae*) diagnosed with fat necrosis. We then compared the expression changes of miRNAs obtained from healthy Korean native cattle of the same age and gender, as well as the affected calf. Subsequently, we performed miRNA-mediated KEGG pathway analysis to identify pathways where miRNAs were significantly upregulated or downregulated.

Fat necrosis in Korean native cattle is a growing concern associated with efforts to increase the fat content in meat. While it is a rare occurrence, its incidence is progressively increasing, leading to economic losses on farms [4]. A study conducted on Japanese black cattle in 2013 reported that around 1000 cattle per year were affected by fat necrosis, resulting in their disposal and further financial implications [18]. Unlike in humans where fat necrosis is not fatal and commonly occurs in the breast, in cattle it primarily affects the abdominal organs, often leading to intestinal obstruction and death [19]. Consequently, fat necrosis in cattle may go unnoticed until rectal palpation or postmortem examination, as is our case. The direct cause of death of the Korean native calf was GDV in conjunction with pronounced hard fat nodule infiltration, leading to serious compression on thoracic as well as peritoneal cavities. In humans, fat necrosis was rarely reported in the neonatal period, namely sclerema neonatorum and adiponecrosis cutis neonatorum. The main feature in both these diseases and in some other affections of fat tissue is crystallization of lipid substances within the vacuoles of fat cells in subcutaneous adipose tissue in humans or in the abdominal and retro-peritoneal fat in animals [7]. Theoretically, crystals of fatty acids and their soaps (hard and firm nodules) might form within fat cell vacuoles because of abnormally swift lipolysis, and hypersaturation of fat. However, it is unknown whether it is related to the age of the individual and species of the animal in question, or absolute or relative insufficiency of enzymes is responsible for the transportation of fatty acids mobilized during lipolysis. It is thought that all affections of adipose tissue with primary crystallization in different species of mammals have the same local origin as a disorder in the metabolism of neutral fats, probably in lipolysis state [5,13]. In humans, the subcutaneous fat has a relatively rapid metabolic turnover. The lipolysis of subcutaneous neutral fat provides to a great extent the energy needed in muscular work. Therefore, if the rate of lipolysis of relatively hypersaturated fat in a newborn or in an infant exceeds the rate of transport of fatty acids in several limited places, adiponecrosis cutis neonatorum may result [13]. In addition, acute or chronic pancreatitis and insulin injection in type 1 diabetes are known factors associated with fat necrosis [20]. While in animals, it has been suggested that acute pancreatitis and tall fescue toxicity may play a role [7,21], the connection with diabetes remains unestablished. Diabetes mellitus in domestic animals has been most commonly reported in dogs and cats but has also been occasionally reported in other species such as cattle, horse, sheep and pigs with no evidence of sex predilection for this disease in cattle [22]. The result of the serum biochemical analysis indicated obesity and pancreatitis in our case, implying a potential association with diabetes. Unfortunately, however, the pancreas was not excavated enough to be tested due to excessive fat infiltration. Therefore, the microarray analysis of miRNAs was applied as the next best thing as promising novel biomarkers for diabetes mellitus, considering their stability and abundance in various body fluids [14,16,17]. Emerging technologies to recover, amplify and detect nucleic acids in the blood have allowed for sensitive methods to make correlations of gene expression profiling with specific states of diseases. Several studies suggest that there is selective expression of circulating miRNA that may correlate with diabetic conditions [16]. However, research on miRNA in cattle is limited, with most studies focusing on milk productivity, pregnancy cycles and infectious diseases [23,24,25,26,27]. Therefore, we referenced research on miRNA related to diabetes mellitus in humans, which is a more actively studied area.

Pancreatic islets regulate glucose homeostasis through insulin and glucagon, which are central to key biological processes and energy homeostasis. Loss or impairment of islet function results in dysregulated blood glucose, which gives rise to multiple life-threatening complications including cardiovascular disease, neuropathy, nephropathy, blindness and stroke. High blood glucose, the most common clinical sign of diabetes, is usually observed only when islet beta cells are already deficient or exhausted [16]. Out of the 783 miRNAs analyzed, 72 miRNAs showed significant differences in expression between cattle with fat necrosis and healthy cattle. Among these, 24 miRNAs exhibited similar expression patterns as observed in patients with type 1 and type 2 diabetes, indicating consistent upregulation or downregulation [14,15,16,17]. However, there were some discrepancies in our findings compared to human studies as well. For instance, the expression levels of bta-miR-22-3p, bta-miR-22-5p and bta-miR-27b, which are typically decreased in humans, were found to be increased in our study [16]. Additionally, miR-375, a well-known miRNA in patients with pancreatitis and diabetes, did not show significant differences compared with that of healthy cattle in our study [28]. On the other hand, the expression levels of bta-miR-26a, bta-miR-29a, bta-miR-30a-5p and bta-miR-181a were consistent with the results of two cohort studies involving juvenile diabetic patients [29]. Furthermore, using TargetScan and Web-Gestalt, we predicted gene expressions associated with the 24 differentially expressed miRNAs. The pathways identified included the prolactin-signaling pathway, insulin resistance, the autophagy pathway, the insulin-signaling pathway, the FoxO-signaling pathway and the PI3K-Akt-signaling pathway. These findings suggest a potential association with type 2 diabetes [30,31,32]. However, the reason for the discovery of cancer-related pathways remains unclear. Overall, this study provides insights into a possible correlation between fat necrosis and diabetes in cattle.

Although diabetes mellitus is rarely reported in cattle, the actual incidence may be higher than reported. In young animals, the development of diabetes can lead to slowed growth, and they are often sold or treated without a specific diagnosis [33]. However, if diabetes develops later in life, there is a better chance of diagnosis in cattle that receive individual therapy. Diabetes in cattle can have various causes and is characterized by pathological changes in the pancreatic tissue. It is often associated with concurrent diseases such as fatty liver, fat cow syndrome, parturition, chronic insulitis and viral diseases, particularly bovine viral diarrhea [34]. Cattle with diabetes experience reduced insulin production in the pancreas, resulting in hyperglycemia, diabetic glycosuria and ketonuria. Diabetes mellitus in cattle is frequently immune-mediated and exhibits similarities to juvenile-onset diabetes mellitus in humans.

## 5. Conclusions

In this study, our main objective was to investigate the potential causes of marked fat necrosis and infiltration on abdominal organs leading to death in a 6-month-old Korean native calf. The lab test results indicated abnormalities in the pancreas, but assessing the pancreas directly was challenging due to extensive damage caused by infiltrated fat in the tissue. Consequently, a microarray analysis of bovine miRNA using serum samples obtained from the calf was performed. These findings indicated that the miRNA analysis can serve as a valuable tool in the investigation of diabetes mellitus in livestock. We observed significant differences in the expression of miRNAs, and the fold changes in these miRNAs resembled those observed in human patients with diabetes mellitus. Furthermore, the predicted KEGG pathways associated with diabetes mellitus were found to be significantly enriched in the calf with fat necrosis. These findings indicate that miRNA analysis can serve as a valuable tool in the investigation of diabetes mellitus in livestock. Our study underscores the importance of exploring the molecular mechanisms underlying fat necrosis and its potential impact on pancreatic function and metabolic disorders in livestock.

## Figures and Tables

**Figure 1 animals-13-02149-f001:**
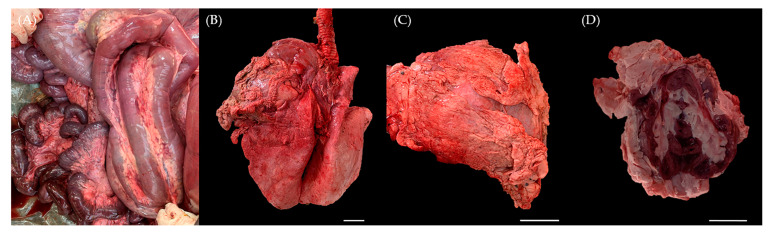
Gross appearance at necropsy. Six-month-old male Korean native calf (*Bos taurus coreanae*). Firm and thick nodular accumulations of fat were located on (**A**) mesentery, (**B**) left cranial pulmonary lobe, (**C**) pericardium and (**D**) kidney. Scale bar = 5 cm.

**Figure 2 animals-13-02149-f002:**
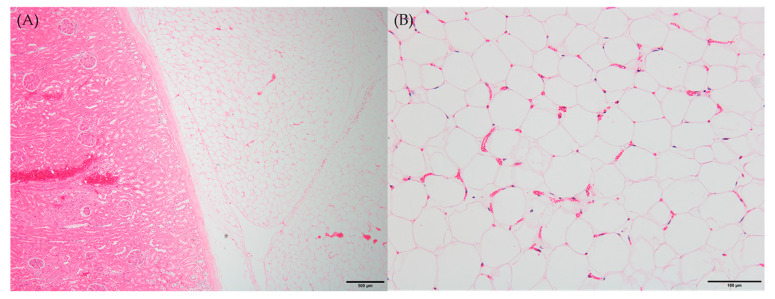
Histopathological findings of necrotic fat tissues. Six-month-old male Korean native calf (*Bos taurus coreanae*). H&E staining. (**A**) Thick fat tissues covered kidney. Scale bar = 500 μm. (**B**) Early stage of fat necrosis. Scale bar = 100 μm.

**Figure 3 animals-13-02149-f003:**
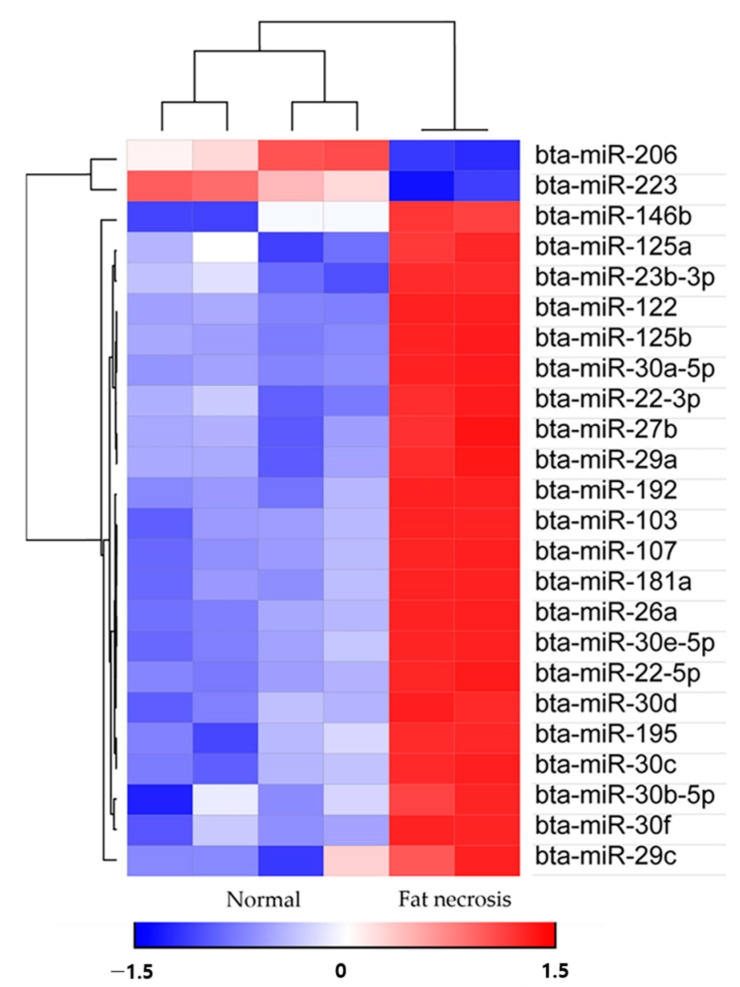
Heat maps with hierarchical clustering of differentially expressed miRNAs between the two groups.

**Table 1 animals-13-02149-t001:** A List of differentially expressed miRNAs associated with diabetes mellitus in a Korean native calf with fat necrosis, along with their corresponding fold changes.

miRNA	Log2 Fold Change	*p*-Value
Upregulation		
bta-miR-22-3p	+3.18	<0.001
bta-miR-22-5p	+3.54	<0.001
bta-miR-23b-3p	+2.03	0.02
bta-miR-26a	+3.35	0.004
bta-miR-27b	+2.16	0.003
bta-miR-29a	+1.97	0.002
bta-miR-29c	+2.18	0.04
bta-miR-30a-5p	+4.32	<0.001
bta-miR-30b-5p	+3.14	0.04
bta-miR-30c	+4.29	0.005
bta-miR-30d	+2.02	0.007
bta-miR-30e-5p	+3.10	0.006
bta-miR-30f	+4.10	0.01
bta-miR-103	+2.71	0.005
bta-miR-107	+2.48	0.004
bta-miR-122	+6.75	<0.001
bta-miR-125a	+3.07	0.03
bta-miR-125b	+4.19	<0.001
bta-miR-146b	+2.46	0.03
bta-miR-181a	+1.83	<0.001
bta-miR-192	+6.35	0.003
bta-miR-195	+6.45	0.02
Downregulation		
bta-miR-206	−4.18	0.04
bta-miR-223	−1.27	0.04

**Table 2 animals-13-02149-t002:** List of significantly enriched KEGG pathways identified in a Korean native calf with fat necrosis.

KEGG Description	Gene Set	Enrichment Ratio	Corrected *p*-Value (FDR)
Prolactin-signaling pathway	bta04917	3.837	0.0092284
Insulin resistance	bta04931	2.886	0.039731
Autophagy	bta04140	2.760	0.033460
Insulin-signaling pathway	bta04910	2.679	0.037608
FoxO-signaling pathway	bta04068	2.582	0.046199
Focal adhesion	bta04510	2.251	0.046199
MicroRNAs in cancer	bta05206	2.223	0.026351
PI3K-Akt-signaling pathway	bta04151	1.968	0.033460
Pathways in cancer	bta05200	1.715	0.045387

## Data Availability

Not applicable.

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
