# Peer review of "A Postmortem Case Study—An Analysis of microRNA Patterns in a Korean Native Male Calf (Bos taurus coreanae) That Died of Fat Necrosis"

_animals, 2023, doi:10.3390/ani13132149_

Round 1
Reviewer 1 Report
Title
- The title is specific and concise but magnifies the results of the study as it alludes to results in cattle, but it is only a case report.
Simple Summary
- The simple summary exceeds the number of words indicated by the journal. (The simple summary consists of no more than 200 words in one paragraph)
- Lines 23-24: it is not methodologically correct to extrapolate results from different contexts and in this case from different species (bovine and human).
- Lines 24 - 31: The summary exaggerates the results of the study as the evidence (serological analysis) does not allow the assumption that the animal had diabetes. Hypertriglyceridemia, hyperamylasemia, hyperlipasemia, and elevated alkaline phosphatase (ALP) levels, described in the results section, relate more to acute pancreatitis (perhaps due to hypertriglyceridemia), probably obstructive, rather than diabetes as such. I recommend the authors to give a new and more appropriate orientation to their manuscript.
Introduction
- The introduction highlights the importance of the study, defines the purpose of the study and its significance, but there is no review of a recent state of research (there are only two publications from the last 5 years). I recommend the authors to make an introduction to the topic with a more updated bibliography.
Material and methods
- The material and methods section is described in sufficient detail. Already known methods are briefly described and appropriately cited.
- Line 137: I am not a statistician, but I wonder if it is possible to assume statistical significance with sample size versus control. “Serum samples were collected from a commissioned bovine and two healthy bovines”.
Results
- The results section provides a concise and precise description of the experiments.
Discussion
- The authors insist on comparing their results with the human species and this is methodologically incorrect. I recommend the authors to focus the discussion of their results with studies ideally conducted in the same species and not to assume diabetes mellitus when there is no evidence for it in their analysis.
Conclusions
- Lines 348 - 349: I recommend the authors to make a proper generalisation of their results and not to conclude with a lack of evidence. “Our analysis yielded interesting findings suggesting a potential link between fat necrosis and diabetes mellitus in the calf.” “These findings indicate that miRNA analysis can serve as a valuable tool in the investigation of diabetes mellitus in livestock”
Author Response
Reviewer #1.
Critique 1. Line 15. Maintain proper gender. e.g. a 6 month-old bull calf is not a cow!
Thank you for the reviewer’s comment. We regret that the expression was inaccurate and has caused confusion. We changed “A 6-month-old Korean native cow” to “A 6-month-old male Korean native calf”.
Critique 2. Lines 24-27, 36-37, 83-84. grammar could be improved.
Thank you for the reviewer’s comment. We checked the grammar error, reviewed it again and did our best to correct grammatical errors including the content. All corrections are reflected in the manuscript in blue font.
Critique 3. Lines 132-33. How accurate are blood samples taken from a cadaver after death? Comments are indicated.
Thank you for the reviewer’s comment. As this comment, the blood samples were taken as soon as possible from a cadaver after being found dead. Because we also recognized the same issue, we measured two types of serum chemistry analysis to check the accuracy and the reproducibility of the results: wet serum chemistry and dry serum chemistry. Both methods showed the same results in the chosen parameters. This is reflected in the manuscript.
Critique 4. No histopath of pancreas is included or lack thereof discussed. No culture of the Abomasun for Clostridiosis is presented or discussed.
Thank you for the reviewer’s comment. We deeply empathize with the importance of histopathologic analysis of pancreas. However, unfortunately, the pancreas was not excavated enough to be tested due to excessive fat infiltration. This was reflected in the manuscript in blue font. Additionally, with all due respect, we find it difficult to confirm the evidence for the comment regarding the culture of abomasum for Clostridiosis. If there was any suspicion of Clostridiosis pathologically, the culture would be proceeded but there wasn’t, rather only physical obstruction of intestinal organs. Thank you for the critical point of view.
Reviewer 2 Report
a. LIne 15. Maintain proper gender. e.g. a 6 month-old bull calf is not a cow!
2. Lines 24-27, 36-37, 83-84. grammar could be improved.
3. Lines 132-33. How accurate are blood samples taken from a cadaver after death? Comments are indicated.
4. No histopath of pancreas is included or lack thereof discussed.
No culture of the Abomasun for Clostridiosis is presented or discussed.
Same as above comments
Author Response
Reviewer #2.
Critique 1. [Title] The title is specific and concise but magnifies the results of the study as it alludes to results in cattle, but it is only a case report.
Thank you for the reviewer’s comment. We so agree with the opinion and modified the title into ‘The analysis of microRNA patterns in one male Korean native calf (Bos taurus coreanae) that perished of fat necrosis’.
Critique 2. [Simple Summary] The simple summary exceeds the number of words indicated by the journal. (The simple summary consists of no more than 200 words in one paragraph)
The simple summary was revised in 200 words according to the reviewer’s comment. Thank you.
Critique 3. [Simple Summary] Lines 23-24: it is not methodologically correct to extrapolate results from different contexts and in this case from different species (bovine and human).
Thank you for the reviewer’s meticulous comment. We reviewed the text again in detail and agree with the reviewer’s comment. However, the sentence was written as such because it was the result of a literature review, and the limitation of our paper were clearly defined in the subsequent linked sentence. It showed a pattern similar to human diabetes, but this case is obviously a diabetic like clinical condition.
Critique 4. [Simple Summary] Lines 24 - 31: The summary exaggerates the results of the study as the evidence (serological analysis) does not allow the assumption that the animal had diabetes. Hypertriglyceridemia, hyperamylasemia, hyperlipasemia, and elevated alkaline phosphatase (ALP) levels, described in the results section, relate more to acute pancreatitis (perhaps due to hypertriglyceridemia), probably obstructive, rather than diabetes as such. I recommend the authors to give a new and more appropriate orientation to their manuscript.
Thank you for the reviewer’s thorough and meticulous comments and time to study our paper. I appreciate it. The sentence describing the serum chemistry result was revised by adding “obstructive acute pancreatitis” after pancreatic damage. The hypothesis for diabetes was originated for investigation of the cause of fat necrosis and the literature review.
Critique 5. [Introduction] The introduction highlights the importance of the study, defines the purpose of the study and its significance, but there is no review of a recent state of research (there are only two publications from the last 5 years). I recommend the authors to make an introduction to the topic with a more updated bibliography.
Thank you for the reviewer’s comment. Unfortunately, recent literatures on non-infectious diseases in large livestock animals such as fat necrosis and diabetes, are limited within the past five years. Related research was conducted extensively from 1970s to 1990s. Although there have been few recent papers on the subject in recent years, metabolic diseases such as fat necrosis continue to occur and cause economic losses, albeit with a low incidence rate. Currently, research results on large animal diseases are scarce due to the shortage of new veterinarians. In this moment, we aim to fulfill its obligation to inform the current and future generations by integrating past research on these diseases with modern analytical approaches. Therefore, we have submitted the manuscript to this journal with the noble purpose of public interest.
Critique 6. [Material and methods] The material and methods section is described in sufficient detail. Already known methods are briefly described and appropriately cited.
Thank you for the reviewer’s comment. I appreciate it.
Critique 7. [Material and methods] Line 137: I am not a statistician, but I wonder if it is possible to assume statistical significance with sample size versus control. “Serum samples were collected from a commissioned bovine and two healthy bovines”.
Thank you for the reviewer’s comment. To perform a statistical analysis, we conducted two replications.
Critique 8. [Results] The results section provides a concise and precise description of the experiments.
Thank you for the reviewer’s comment.
Critique 9. [Discussion] The authors insist on comparing their results with the human species and this is methodologically incorrect. I recommend the authors to focus the discussion of their results with studies ideally conducted in the same species and not to assume diabetes mellitus when there is no evidence for it in their analysis.
Thank you for the reviewer’s comment. With all due respect, we did not intend to extrapolate results. We have provided evidence through literature review that the significant changes observed in miRNA expression in the calf exhibit similar patterns to those reported in humans in terms of the types of miRNA and the extent of increase or decrease. The limitation of our paper were clearly defined as a diabetic like clinical condition.
Critique 10. [Conclusions] Lines 348 - 349: I recommend the authors to make a proper generalisation of their results and not to conclude with a lack of evidence. “Our analysis yielded interesting findings suggesting a potential link between fat necrosis and diabetes mellitus in the calf.” “These findings indicate that miRNA analysis can serve as a valuable tool in the investigation of diabetes mellitus in livestock”
Thank you for the reviewer’s kind suggestion. The sentence has been corrected as you kindly suggested and further developed. Thank you.